# A Two-Stage Framework for Mathematical Expression Recognition

## Abstract

Although mathematical expressions (MEs) recognition have achieved great progress, the development of MEs recognition in real scenes is still unsatisfactory. Inspired by the recent work of neutral network, this paper proposes a novel two-stage approach which takes a printed mathematical expression image as input and generates LaTeX sequence as output. In the first stage, this method locates and recognizes the math symbols of input image by object detection algorithm. In the second stage, it translates math symbols with position information into LaTeX sequences by seq2seq model equipped with attention mechanism. In particular, the detection of mathematical symbols and the structural analysis of mathematical formulas are carried out separately in two steps, which effectively improves the recognition accuracy and enhances the generalization ability. The experiment demonstrates that the two-stage method significantly outperforms the end-to-end method. Especially, the ExpRate(expression recognition rate) of our model is 74.1%, 20.3 percentage points higher than that of the end-to-end model on the test data that doesn't come from the same source as training data.

## 1 Introduction

Mathematical expressions (MEs) play an essential role in math, physics and many other fields. Recognizing mathematical expressions is receiving increasing attentions for application in digitization and retrieval of printed documents. The process of recognizing mathematical expressions is to convert mathematical expressions into LaTeX strings, which includes three stages: symbol segmentation, symbol recognition and structural analysis. We usually divide recognition of MEs into handwritten and printed domains. In the domain of printed MEs, researchers face three challenges(Anderson (1967); Belaid & Haton (1984)): the complicated two-dimensional structures, various styles of images in printed input and strong dependency on contextual information.

Three major problems are involved in MEs recognition (Zanibbi et al. (2012); Mouchre et al. (2016)): symbol segmentation, symbol recognition, structural analysis. These problems can be solved sequentially or globally. Deng et al. (2016a) proposed an end-to-end formula recognition method for images generated *directly from LaTeX code*. For their method, a CNN is applied to extract visual features from the input images and then every row is encoded using a recurrent neural network (RNN). These encoded features are then used by an RNN decoder with a visual attention mechanism to produce final outputs.

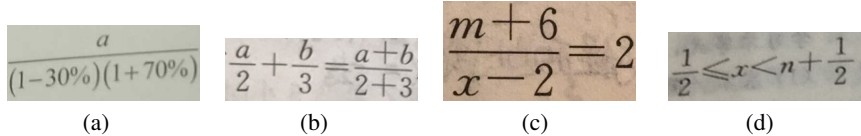

(a)          (b)          (c)          (d)

Figure 1: ME images from real world

Their model achieved 77.46% accuracy on test data which had the same distribution as the training data. For convenience, we define *homologous test data* and *non-homologous test data* here. It is called *homologous test data* if the test data comes from the same source as the training data.

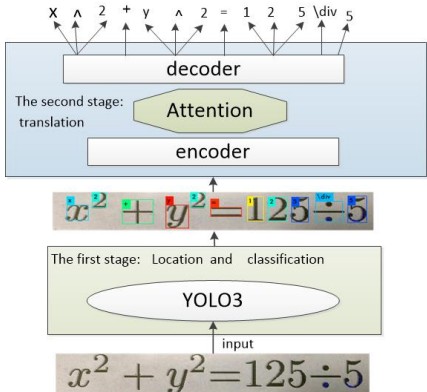

Figure 2: Two-stage method: the first stage is to use YOLOv3 to detect the symbols, the second to translate the symbols with position information into LaTeX strings by a seq2seq model.

Otherwise, It is called *non-homologous test data*. The result demonstrated that the model had good performance on homologous test data. However, in practice we found that the model had poor performance on non-homologous test data, that is, the generalization ability of the above model is very weak. In real scenes, the images may be of great variety, like a wide range of backgrounds, as is shown in Figure 1. It is impossible to predict all input cases and the test data may diverge from what the system has seen before. Therefore, the method above is not appropriate to be applied to recognize images from real scenes. It seems a necessary task to design a model that has strong generalization ability to recognize real MEs images.

To improve the generalization ability, we decouple the feature extraction process and the translation process. In the feature extraction process, we use YOLOv3(Redmon & Farhadi, 2018) to locate and classify the symbols of images. Then the class and location information of each symbol is vectorized, which is used as input for the seq2seq model and translated into LaTeX strings.

The two-stage approach has several benefits. Firstly, changes in the input images styles would have no influence on the encoder-decoder model, and YOLOv3 has good generalization ability(Redmon et al., 2015; Redmon & Farhadi, 2018), so the two-stage method would have better generalization ability than the end-to-end method(Deng et al., 2016b). Secondly, the feature vectors composed by position and classification information are much more concise than the feature vectors extracted directly by the convolutional layers(Deng et al., 2016b), which are easier to learn and get higher recognition rate.

The main contributions of this paper can be summarized as:

(1) A two-stage method for MEs recognition is proposed to decouple the feature extraction process and the translation process, which has better generalization ability and achieve better accuracy.

(2) By concatenating position information and classification information into feature vectors, we successfully translate symbols with position information into LaTeX strings by the seq2seq model. To the best of our knowledge, we use the seq2seq model to solve structural analysis problem for the first time and achieve satisfactory results, which may accelerate progress in machine recognition of other two-dimensional languages.

(3) We propose a method to automatically generate MEs images with position and classification information of the symbols, which avoid expensive manual annotation.

## 2 RELATED WORK

The detection framework based on state-of-the-art convolutional neural network (CNN) can be divided into two categories: one-stage method and two-stage method. The typical two-stage methods include R-CNN(Girshick et al. (2014)), Fast R-CNN(Uijlings et al. (2013)), Faster RCNN (Ren et al.

(2017)) and R-FCN(Dai et al. (2016)), which firstly utilize the region proposal generation algorithms to generate potential symbol regions and then perform classification on the proposed regions. These methods improve the accuracy, but yield slow processing speed. On the contrary, single-stage methods like SSD(Liu et al. (2016)) and YOLO(Redmon et al. (2015)), apply predefined sliding default boxes of different scales on one or multiple feature maps, thus keep balance between speed and accuracy.

Aiming to structural analysis, some papers prefer approaches based on predefined grammars to solve the problem. For example, Lavirotte (1998) investigated graph grammar and graph rewriting as a solution to recognize two dimensional mathematical notations. Chan & Yeung (2001) developed a system based on definite clause grammars(DCG), which incorporated an error detection and correction mechanism into a parser. Huang (2007) proposed a structural analysis approach based on the Attribute String Grammar and the Baseline Tree Transformation approaches. lvaro et al. (2014) introduced hidden Markov models to recognize mathematical symbols and stochastic context-free grammar to model the relationship between these symbols. Yamamoto et al. (2006) et al. proposed a new two-dimensional structure model for mathematical expressions by the new concept of Hidden Writing Area (HWA). Maclean & Labahn (2015) presented a system, which captured all recognizable interpretations of the input and organized them in parse trees by a Bayesian scoring model. lvaro et al. (2016) described a statistical framework of grammar-based approach that deal with structured problems by two-dimensional grammars.

Some papers introduced an end-to-end (Tao et al. (2013); Cirean et al. (2010); Deng et al. (2016a)) framework totally based on neutral networks for mathematical expressions which did not require a predefined math grammar. Zhang et al. (2017b) proposed a model namely Watch, Attend and Parse (WAP), which had two components: a watcher and a parser. In Zhang et al. (2017a), the authors improved the CNN encoder by employing a novel architecture called densely connected convolutional networks (DenseNet)(Gao et al. (2016)).

## 3 NETWORK ARCHITECTURE

In this section, we will give a detailed introduction about the two-stage framework proposed in this paper for printed mathematical expressions recognition. In the first stage, the main task is to locate and recognize the math symbols of input images by YOLOv3. In the second stage, the task is to translate the vector containing symbols categories and positional information into LaTeX strings. We will elaborate on the classic attention mechanism based on encoder-decoder framework and explain the process of parsing. As illustrated in Fig. 2, the detection is YOLOv3, which generates a series of symbol sequences with location information from the whole MEs images. Then these information including symbols' location and classes are processed and converted into vectors as input to the seq2seq model. Finally, the output of the decoder is LaTeX strings.

### 3.1 DETECTION:YOLO3

Symbol detection in formula images are more complex than other natural scene object detection due to its smaller contours and various shapes. There are many small symbols, like '·,, '−', and subscript and superscript symbols, which may be lossed in symbol detection. Besides, some symbols are very similar, like digit '1' and letter 'l', which might cause error classification. Error detection will seriously affect the parsing in next stage, thus reduce the final formula recognition accuracy.

Among many object detection algorithms, YOLOv3 has proved to be the most effective in detecting small objects. YOLOv3 has integrated advanced conception from other object detection algorithms like feature pyramid networks (Lin et al., 2016) and anchor box prediction mechanism (Redmon & Farhadi, 2017). It uses entire image as input of the network and draknet-53 without fully connected layer to extract feature, and finally predicts the bounding box of object and its category directly at the output layer. In the part of predicting boxes, it extracts features from three different scales and merges up-sampled feature map with earlier feature map, thus get better performance in small object detection.

We adjust the size of the anchor box to reduce missed detection. Especially, anchors like (10,10), (6,40) and (40,6) are added to detect points, horizontal or vertical lines. For similar symbols like

digit '1' and letter 'l', it is almost impossible to be distinguished by the model. Such errors might be corrected if extra rules are added as a postprocessing.

## 3.2 PARSE:ENCODER-DECODER

A lot of grammar-based parsing algorithms have been proposed for structural analysis and performed well in several systems (lvaro et al. (2014); Awal et al. (2014); lvaro et al. (2016)). But grammar-based methods requires priori knowledge to define ME grammar and its complexity increases exponentially with the size of the predefined grammar.

The encoder-decoder framework was first presented in the paper(Cho et al. (2014)), which was used to translate one language into another. Structural analysis is to parse symbols with position information into LaTeX sequence, which is similar to translation. The only difference between them is that the former is two-dimensional and the latter is one-dimensional. By creatively concatenating position information and classification information as input vectors, we successfully employed encoder-decoder model to perform structural analysis of MEs.

### 3.2.1 ENCODER-DECODER

The LSTM is an improved version of simple RNN, so we employ LSTM as the encoder for alleviating the vanishing and exploding gradient problems(Bengio et al. (2002)). In encoding part, the input of each time step is a symbol with position information, represented by feature vector $(x, y, w, h, \textbf{o-h})$ where $(x, y, w, h)$ are the coordinates of the top-left corner, width and height of the rectangle that contains the symbol, and **o-h** is a one-hot vector to distinguish the symbol from others.

Similarly, we employ lstm as the encoder. In the decoding stage, we aim to generate a most likely LaTeX string given the input feature vectors:

$$\hat{y} = \arg\max \log P(y|x) \tag{1}$$

In fact, the Y sequence is closely related to the semantic vector C generated by the encoder, and $y_t$ at one time also is affected by y generated at other times, which can be described as the following formula:

$$y_t = \text{argmax} P(y_t) = \prod_{t=1}^{T} p(y_t | \{y_1, y_2, \dots, y_{t-1}\}, C) \tag{2}$$

In addition, it is intuitive to adopt the ensemble method(Dietterich (2000)) for improving the performance of beam algorithm.

### 3.2.2 ATTENTION

Although, the encoder-decoder model has been widely applied to many problems, its biggest limitation is that the only link between encoding and decoding is a fixed-length semantic vector C. However, this semantic vector may not fully represent the information of the whole sequence, and the information carried by the first input will be diluted by the later input information. The accuracy of model is also affected by inadequate input sequence information. To solve this problem, Bahdanau et al. (2014) proposed attention model that generates an "attention range" when output is generated, indicating which parts of the input sequence should be paid attention to in the next output, and then generates the next output according to the region of interest.

Between encoder and decoder, we introduce attention mechanism. In standard attention(Bahdanau et al. (2014)), we compute a context vector candidate $\hat{c}_i$ as:

$$\hat{c}_i = \sum_{j=1}^{T_x} \alpha_{ij} h_j \tag{3}$$

Where $h_j$ is the hidden state of encoder and $\alpha_{ij}$ is the correlation coefficient. We use a neural network to approximate the attention distribution $\alpha_{ij}$:

$$\alpha_{ij} = \mathrm{softmax}\,(u_t) = \frac{\exp{(e_{ti})}}{\sum_{k=1}^{L} \exp{(e_{tk})}} \tag{4}$$
$$e_{ti} = v_{att}^{T} \tanh{(W_{att}h_t + U_{att}a_i)}$$

Where $e_{ti}$ denotes the energy of annotation vector $a_i$ at time step $t$ conditioned on the current lstm hidden state prediction $h_t$.

## 4 EXPERIMENTS

To validate the effectiveness of the proposed method for printed mathematical expression recognition, we design a set of experiments to evaluate the answers of the following questions:

Q1 Is the improved YOLOv3 for locating and recognizing mathematical symbols effective?

Q2 How does encoder-decoder analyze the 2D structure of MEs?

Q3 Does the proposed approach outperform state-of-the-arts?

### 4.1 METRIC

**Metric of detection model:** For detection tasks, IoU (Intersection-over-Union) is a standard metric to measure performance. In this paper, detections are considered true or false positives based on IoU. A detection is to be considered correct if the IoU exceeds 50% and the classification is right.

**Metric of translation model:** For the translation model, we use accuracy and Word Error Rate(WER)Klakow & Peters (2002) as metrics to measure the performance. The accuracy is calculated based on expression recognition rates (ExpRate), i.e., the percentage of predicted mathematical expressions *exactly* matching the ground truth, which gives a useful global performance metric.

### 4.2 DATA

Data with classification and position information is expensive to annotate and rare. To address this problem, a tool is developed to provide detailed ground-truth annotations, which are cheap and more precise than data annotated manually. After filtering some inappropriate equations IM2LATEX-100K (Deng et al., 2016b), a data set containing 81214 equations is obtained to generate mathematical expression images. Besides, 6000 background images are collected and split into two parts: one part contains 4000 images, which are used as background of training set, validation set and the first test set; the other part 2000 images are used as background of the second test set. The two test sets are generated by the same LaTeX equations but with different background. The first test data set, called *homologous test set*, has the same distribution as the training set. The second test set, called *non-homologous test set*, doesn't have the same distribution as the training set, which is used to evaluate the generalization ability of the models. For details to generate data, see Appendix A.

### 4.3 DETECTION MODEL EXPERIMENTS

We train the model using minibatch SGD with four GPUs, 0.9 momentum, 0.0005 weight decay, and batch size 64. We first train the model with 0.0005 learning rate for 20000 iterations, and then continue training for 4000 iterations with 0.0001 and 1000 iterations with 0.00001. The anchors we use are (10,10), (6,40), (40,40), (40,6), (80,80), (12,100), (120,120), (100,12), (228,228). Especially, anchor (10,10) is designed to detect small symbol like '·' and '.', and the anchors (6,40), (40,6), (12,100), (100,12) to detect horizontal or vertical lines, like '-', '—' or '|'.

Figure 3 gives an example of YOLOv3. Testing on 5000 images of the *homologous test set*, there are 4822 images that all the symbols are correctly located and recognized. Besides, location errors occurs in 125 images and classification errors occur in 62 images, among which there are 9 images that contain both location and classification errors.Testing on 5000 images of the *non-homologous test set*, there are 4813 images that all the symbols are correctly located and recognized. In addition,

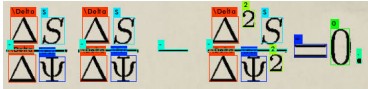

Figure 3: An example of YOLOv3.

location errors and classification errors occur in 126 images and 72 images, respectively. The images that contain both location and classification errors are 11.

Table 1 shows the testing results of both homologous and non-homologous test set, which are almost identical. Although the detection model has never 'seen' the background images of the non-homologous test set, it still achieves good performance. This demonstrates the detection model has good generalization ability, which are very important in practical applications.

Table 1: Testing result on homologous/non-homologous test set

| Test Set | NO Error | Loc Errors | Cls Errors | Loc and Cls Errors |
|---|---|---|---|---|
| Homologous | 96.44% | 2.5% | 1.24% | 0.18% |
| Non-homologous | 96.26% | 2.52% | 1.44% | 0.22% |

## 4.4 TRANSLATION MODEL EXPERIMENTS

We regard the conversion of symbol sequences with location information into LaTeX sequences as a translation process and solve this problem with the encoder-decoder model, which is of great significance. As far as we know, we are the first to use the encoder-decoder model to solve the two-dimensional translation problem. As is shown in Section 4.2, we have 81214 different LaTeX math equations, 71214 expressions for training, 5000 for validation and 5000 for test. In order to train the encoder-decoder model, we run our YOLOv3 model on the training dataset, validation dataset and test dataset, respectively and store the symbols' position and classification information of every image. After filtering the wrong cases, we obtained 69405 training expressions, 4826 validation expressions and 4822 test expressions.

The encoder is a bi-directional LSTM. The hidden state of the decoder is of size 512, encoder LSTM of 256, which are the same as the configuration of the RNN layers in Im2LaTex (Deng et al., 2016b). The model was trained using SGD with batch size 64 for 120 epochs. The initial learning rate is 0.001, and reduced to 0.0001 after 100 epoches. Besides, the input symbols are ordered by the upper-left corners of their boxes from left to right, top to bottom. For example, the input order of the symbols is -, m, k, c, 2, =, $\sqrt{}$, 3, T for expression $\frac{mc^2}{k} = \sqrt{3}T$.

Since we have only 69405 mathematical expressions to train the encoder-decoder model, how to avoid overfitting is a great challenge. To address the problem, we use extensive data augmentation. Firstly, we replace symbols with other symbols to get more mathematical expressions. For example, we replace the 'a', 'b', 'c' and '2' in $a^2 + b^2 = c^2$ with four other random chosen symbols 'e', 'g', 'l' and '4', thus obtain a new equation $e^4 + g^4 = l^4$. For position information, we introduce random scaling and translations. For each training epoch, up to 80% of the 69405 mathematical expressions would be augmented by the strategy above.

*Data augmentation* plays an important role in training the encoder-decoder model. The experiment shows that we can improve12.8% ExpRate with the augmentation strategy above. Why does the encoder-decoder model benefit so much from the augmentation strategy? One reason may be that the data is *not enough* to train a good seq2seq model and the feature vectors(position + one-hot vectors) are *too concise*, which don't contain any noise compared to the feature vectors extracted by the convolution layers in Im2LaTex (Deng et al., 2016b).

In Table 2, the performance of standard attention model is clear to be observed. For homologous test set, the ExpRate is 78.0% and the wer is 4.4%. Besides, the result of non-homologous test set is almost the same as that of homologous test set. It is understandable because the input of the encoder-

Table 2: Testing result of the encoder-decoder model

| Test Set | WER | ExpRate | $\leq 1\%$ | $\leq 2\%$ |
|---|---|---|---|---|
| Homologous | 0.044 | 78.0% | 83.1% | 85.9% |
| Non-homologous | 0.045 | 77.9% | 83.1% | 85.7% |

decoder model is the position and classification information of symbols, not the image. Obviously, changes in image style has little influences on the encoder-decoder model.

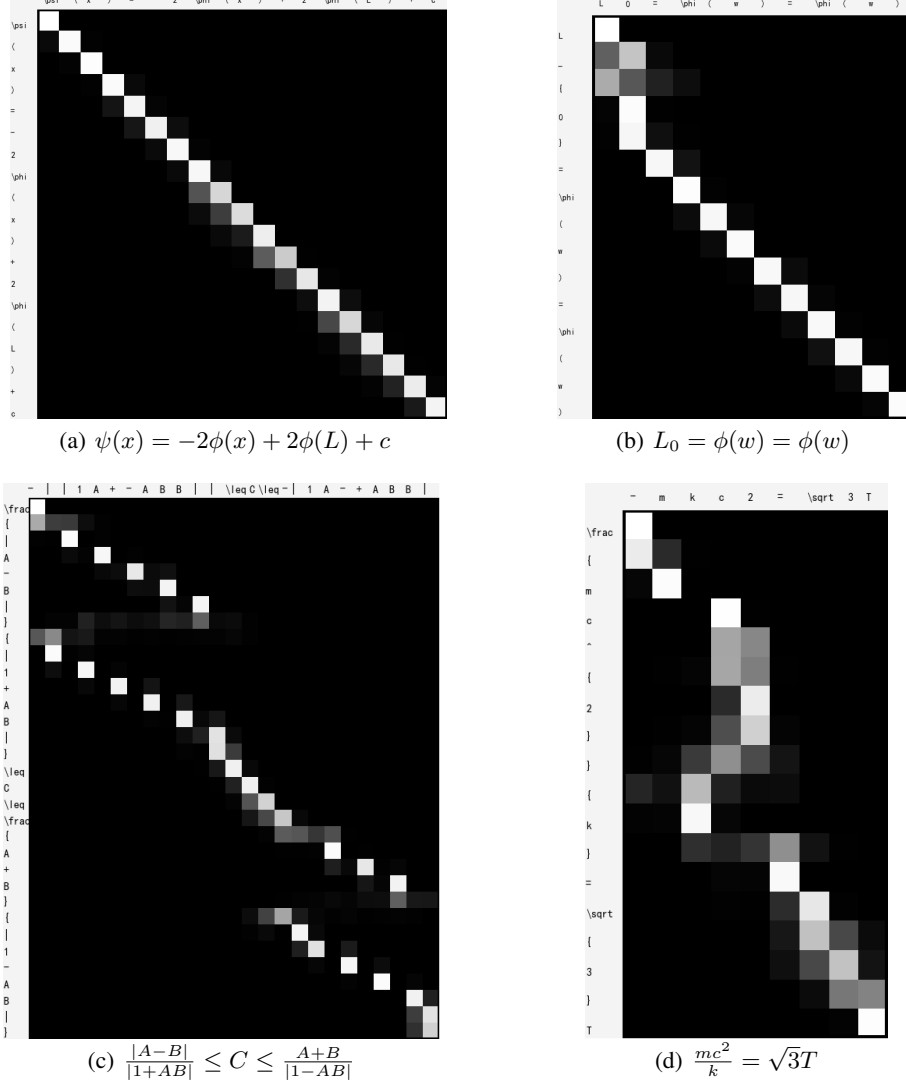

(a) $\psi(x) = -2\phi(x) + 2\phi(L) + c$

(b) $L_0 = \phi(w) = \phi(w)$

(c) $\frac{|A-B|}{|1+AB|} \leq C \leq \frac{A+B}{|1-AB|}$

(d) $\frac{mc^2}{k} = \sqrt{3}T$

Figure 4: Four sample alignments by the encoder-decoder model. The x-axis and y-axis of each plot are the symbols with position information and the generated translation (LaTeX), respectively. Note that we omit the position information here. Each pixel shows the weight $\alpha_{ij}$ of the annotation of the $j$-th source word for the $i$-th target word, in grayscale (0: black, 1: white)

We would inspect the alignment between the symbols (with position information) in a source sentence and the words in a generated LaTeX by visualizing the annotation weights $\alpha_{ij}$ from Eq. (4), as in Fig. 4. Each row of a matrix in each plot indicates the weights associated with the annota-

tions. From the matrix we see which positions in the source sentence were more important when generating the target LaTeX.

From the Fig. 4 we can conclude that the alignment between the symbols with position information and the target LaTeX is monotonic. In general, weights along the diagonal of each matrix are closer to 1. However, there are also many non-trivial, non-monotonic alignments. It's obvious that the alignments depend on the spatial relationships between the mathematical symbols, which might be horizontal, vertical, subscript, superscript or inside. Fig. 4(a) shows a strictly monotonic alignment since all the symbols are horizonal relationship. Vertical, subscript, superscript or inside relationships are usually ordered differently between the input symbols and generated LaTeX. When a target LaTeX contains several different spatial relationships, like $\frac{mc^2}{k}$ (vertical and superscript), the weights seem disorderly in some degree. In Fig. 4(d), the symbol 'k' of the input sequence is ahead of the symbol 'c', but the encoder-decoder model was able to correctly align the symbol 'c' with numerator and the symbol 'k' with denominator, which show that the model successfully captured the position information of the symbols. Fig. 4(c),(d) demonstrated the encoder-decoder model can handle complicated two-dimensional structure.

## 4.5 EVALUATE AS A WHOLE

The current state-of-art OCR-based mathematical expression recognition system is Im2LaTex(Deng et al., 2016b), an end-to-end recognition system, which combines symbol recognition and structural analysis as a whole. In this part we run experiments comparing our model to Im2LaTex. To make a fair comparison, we downloaded the code they exposed in GitHub and trained with the same data set.

Table 3: Testing result of the homologous test dataset

| Model | WER | ExpRate | $\leq 1\%$ | $\leq 2\%$ |
|---|---|---|---|---|
| Ours | 0.048 | 74.3% | 79.7% | 82.8% |
| Im2LaTex | 0.054 | 73.37% | 78.5% | 80.9% |

Table 4: Testing result of the non-homologous test dataset

| Model | WER | ExpRate | $\leq 1\%$ | $\leq 2\%$ |
|---|---|---|---|---|
| Ours | 0.050 | 74.1% | 78.21% | 81.5% |
| Im2LaTex | 0.1355 | 53.9% | 60.9% | 64.56% |

In Table 3, we can observe that our model achieved an ExpRate of 74.3%, while its WER was only 0.048 for homologous test set, which is slightly better than Im2LaTex. However, Table 4 shows the results of the non-homologous test dataset. The ExpRate of our model is 74.1%, 20.3 percentage points higher than Im2LaTex. The $\leq 1\%$ error percentages ( 78.21%) and $\leq 2\%$ error percentages ( 81.5%) of our model is also much higer than Im2LaTex. From these indicators, our system outperforms Im2LaTex for homologous test dataset, and is evidently better than Im2LaTex for non-homologous test dataset. So our model has better generalization, which is superior to Im2LaTex for recognizing MEs with real backgrounds.

## 5 CONCLUSIONS

In this study, we introduce a two-stage framework to recognize mathematic expressions. It is the first work that employs YOLOv3 to detect mathematic symbols and uses the seq2seq model based attention to perform structural analysis. We demonstrate through experiment results that the novel two-stage framework has better generalization ability and performs better than the state-of-the-art methods in real scenario. In future work, we plan to further improve the accuracy of parser by adding more data and introducing more advanced mechanisms, thereby improving the accuracy of the whole mathematical formula recognition system.

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

## A  DATA

### A.1  BACKGROUND IMAGES

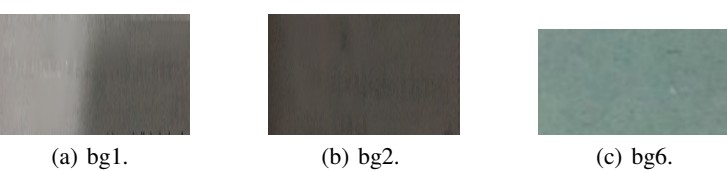

(a) bg1.          (b) bg2.          (c) bg6.

Figure 5: Background images

To favour variety, 6,000 background images are collected from printed articles, books, examination papers and so on. We collected a lot of printed articles, books, examination papers and then took photos of them. We only cropped the non-text areas of the photos as background images. Figure 5 shows some examples of the background images.

## A.2 DATA GENERATION

There are two kinds of mathematical symbols: single-component (like 'a', 'b', 'c', etc) symbols and multi-components symbols (like 'i', 'j', '=', etc). To get the position information, the multi-components symbols are rendered with specific colors. Then the single-component symbols are segmented by connected components and the multi-components symbols by colors. After obtaining the position information, the symbols were cropped and then classified by a neural network. Eventually, we obtained a dataset with 81214 images and corresponding location and category information of symbols. Finally, the images are binarized and blended into the background images using Poisson image editing(Rez et al., 2003). Figure 6 gives an example to show the process to generate data.

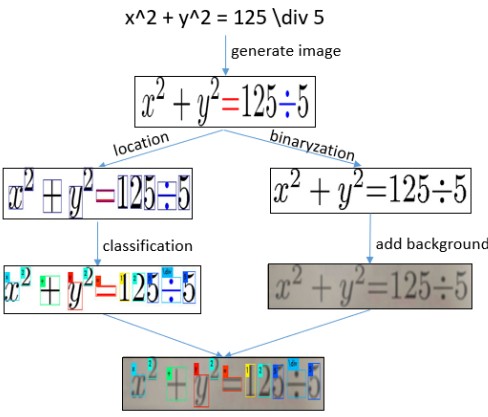

Figure 6: The process to generate data

