# OpenReview forum: "A TWO-STAGE FRAMEWORK FOR MATHEMATICAL EXPRESSION RECOGNITION"
_ICLR.cc/2020/Conference — Reject_

### Official Review · AnonReviewer1 · 2019-10-23
**Official Blind Review #1**

**Rating:** 1

**Review:**

This paper considers the problem of converting an image of a math expression into LaTeX.  They note that while the model proposed in Deng et al works well on the IM2LATEX-100K dataset, is doesn't generalize well to equations in real-world settings that you'd have in a photograph or a scan of an equation.  They propose an approach that breaks the problem into two steps.  In the first step they detect all the characters in the image, identifying the  character type and bounding box for each.  In the second step they use an encoder/decoder (LSTM/LSTM with attention) model to translate this sequence of character encodings into a LaTeX sequence.  They create a new dataset of LaTeX equations rendered on backgrounds sampled from real photographs of books and papers.  They split this into a training and "homologous" test set.   They find performance of their two stage model is a bit better than Deng's model on this test set.  They then create a "non-homologous" test set, in which the test equations are rendered on a new set of backgrounds, unseen in training.  On this test set, the model in the paper performs essentially the same as on the homologous dataset, while the Deng model performs substantially worse.  The conclusion is that the two stage approach creates a model that is much more robust to the appearance of the equation.

I think the main takeaway from this paper is that there can be disadvantages to an end-to-end approach.  To achieve their improvement, authors used their insight that there is an intermediate representation that summarizes all relevant information (the sequence of characters and positions), together with the ability to generate a new training set automatically to learn this intermediate representation.

I think this is an interesting case study in applied machine learning, but I don't think it will be of enough general use or interest to the ICLR community to merit acceptance.

As an applications paper, I think there are several aspects that can be improved.  Here are some specific questions and comments that may help a further iteration of this paper:
- You have examples of ME images from the real world, but you don't have any examples of your artificial "real world" equations, overlayed on your sampled backgrounds.  Those would be helpful. Are any other modifications done to the equation to simulate a real-life picture or scan, such as color or darkness distortions, angles, etc?
- How does your proposed model perform on the original IM2LATEX-100K problem?
- How well does the encoder-decoder model do an a gold encoding of the input?  It would be nice to separate the errors into translation errors vs object detection errors.
- Can you give examples of scenarios where your model got things right and Deng's model did not, and vice versa?  Is there any interpretation to why each model does better or worse for various example?  I imagine the model may have more difficulty with equations where one needs to refer to characters that are on the left of the image quite late in the LaTeX expression.  For example, fractions with long numerator expressions and and cases environments.
- Did you try or consider using a soft encoding of the character identity, instead of a one-hot?  Perhaps there are context clues that the encoder/decoder model could use to disambiguate between a 1 and an l, for example.


**Experience Assessment:**

I have published one or two papers in this area.

**Review Assessment: Checking Correctness Of Derivations And Theory:**

N/A

**Review Assessment: Checking Correctness Of Experiments:**

I assessed the sensibility of the experiments.

**Review Assessment: Thoroughness In Paper Reading:**

I read the paper thoroughly.

---

### Official Review · AnonReviewer3 · 2019-11-01
**Official Blind Review #3**

**Rating:** 3

**Review:**

In this paper the authors propose a two stage pipeline that aims to solve for mathematical expression recognition. The main approach uses the following stages, a detection stage that is based on YoloV3 and a sequence to sequence approach. The authors compare their method against Image2Latex approach (2016) that is an end to end pipeline and show that there is significant improvement compared to this approach.

However, this problem has been a standard task and solved both in the handwritten math expression problem (CROHME challenge of ICDAR and typeset formula detection and recognition. There have been much progress through these challenges with various teams competing. A variety of approaches have been tried for this task and unfortunately the present work has not compared nor evaluated against these approaches.

Im2Latex work is quite old benchmark and there have been numerous works as have been cited by the authors and more as can be available from the challenge. The methods presented are also not novel. Using Yolov3 for detection and sequence2sequence for parsing expressions are more or less standard approaches. Hence, the proposed work does not add a significant insight in solving the problem.

**Experience Assessment:**

I have read many papers in this area.

**Review Assessment: Checking Correctness Of Derivations And Theory:**

I assessed the sensibility of the derivations and theory.

**Review Assessment: Checking Correctness Of Experiments:**

I assessed the sensibility of the experiments.

**Review Assessment: Thoroughness In Paper Reading:**

I read the paper at least twice and used my best judgement in assessing the paper.

---

### Official Review · AnonReviewer2 · 2019-11-01
**Official Blind Review #2**

**Rating:** 6

**Review:**

This paper proposes to recognise a mathmatical expression using a two-stage framework, including object detection by YOLOv3, and encoder-decoder based translation. The paper is written well, and easy to read. In the experiments, the recognition and translation methods both work well on Homologous and non-Homologous test data. In particular, the proposed method improved the performance over the state-of-the-art method Im2LaTex. Addtionally, the authors visualised the sample alignments of encoder-decoder model, which is helpful for understanding the method.

A few comments are as follow:
1) On Page 1, the first and second paragraphs both contain "symbol segmentation, symbol recognition and structural analysis". It looks this framework is repeated again and again.
2) In formula (4), the reviewer did not see the explanation of "v_{att}^T" and "u_T".
3) Some references have only authors and title information, without conference/journal information.


**Experience Assessment:**

I have read many papers in this area.

**Review Assessment: Checking Correctness Of Derivations And Theory:**

I assessed the sensibility of the derivations and theory.

**Review Assessment: Checking Correctness Of Experiments:**

I assessed the sensibility of the experiments.

**Review Assessment: Thoroughness In Paper Reading:**

I read the paper at least twice and used my best judgement in assessing the paper.

---

### Decision · Program_Chairs · 2019-12-19

**Decision:**

Reject

**Comment:**

One reviewer is positive, while the others recommend rejection. The authors did not submit a rebuttal, thus the reviewers kept their original assessment.